# Tolerance of Placozoa for temperate climates: Evidence for known and new placozoan clades in the southern waters of Australia

Bree A. Wright[1], Hans-Jürgen Osigus[2]*, Moritz J. Schmidt[2]*, Julian Ratcliffe[3]*, Kai Kamm[2]*, Gabriela C. Martinez-Ortiz[4]‡, Martina Rehn[5]‡, Marc Kvansakul[1]*, Bernd Schierwater[2]*, Patrick O. Humbert[1,6,7]*

**1** Department of Biochemistry & Chemistry, La Trobe Institute for Molecular Science, La Trobe University, Melbourne, Victoria, Australia, **2** Institute of Animal Ecology & Evolution, University of Veterinary Medicine Hannover, Foundation, Hannover, Germany, **3** La Trobe University Bioimaging Platform, La Trobe University, Melbourne, Victoria, Australia, **4** Department of Energy, Environment and Climate Action, Victoria State Government, Melbourne, Victoria, Australia, **5** La Trobe University, Melbourne, Victoria, Australia, **6** Department of Biochemistry & Pharmacology, University of Melbourne, Melbourne, Victoria, Australia, **7** Department of Clinical Pathology, University of Melbourne, Melbourne, Victoria, Australia

☙ These authors contributed equally to this work.
‡ GCM-O and MR authors also contributed equally to this work.
* p.humbert@latrobe.edu.au (POH); bernado@trichoplax.com (BS); m.kvansakul@latrobe.edu.au (MK)

## Abstract

Placozoans are small multicellular sea-dwelling animals that are typically found in shallow, warm ocean waters and have been reported in various marine environments worldwide. Their unique morphology makes them a powerful new model organism to study the evolutionary cell biology in early metazoans. Yet, knowledge on their biodiversity and ecological distribution is severely limited. Here, we report the isolation of placozoans in the temperate waters of Victoria, Australia, their most southern location known to date. Using light, electron, and confocal microscopy, we describe their morphology and behaviour. In addition to the known haplotypes H2 and H17, we have identified a new haplotype, here designated as H20, which defines a new placozoan clade. This study provides novel insights into the distribution, ecological niche separation and genetic diversity of placozoans, and reports the first morphological and ultrastructural characterisation of placozoan clades isolated from the southern waters of Victoria, Australia.

## Introduction

Placozoans are millimetre sized animals that are found globally in oceanic waters [1,2]. They are among the earliest branching lineages in the animal kingdom and are considered the structurally simplest of all metazoans [3,4]. These properties have attracted considerable interest from the scientific community for their use as a simple model organism to study early metazoan evolution at a cellular level, including research on cell death, innate immunity, gravitational biology and cancer [2,5–9]. Placozoans have at least nine identifiable major somatic cell types [4,10] and can be described as tri-layered animals consisting of an upper epithelium,

**Data availability statement:** All relevant data are within the manuscript and its Supporting Information files.

**Funding:** POH,MK La Trobe Univerisy Grant Ready Scheme funding no.2000004451 Funding body had no participation in the preparation or completion of any work.

a highly ciliated lower epithelium and fibre cells that are dispersed throughout the middle of the animal [11–13]. Placozoans typically comprise of approximately 20,000 cells, of which 80% are epithelial cells [13]. However, placozoans are known to vary significantly in size from under 1 mm to over 5 mm [14,15], with the exception of *Polyplacotoma mediterranea*, which can reach a size of more than 10 mm [16]. While sexual reproduction has occasionally been observed in the laboratory, the complete lifecycle of placozoans still remains unknown [12]. In the lab they predominantly reproduce by fission and no juvenile states have been identified neither in the field nor under laboratory conditions [2,10,17]. These tiny creatures are abundant in warm to temperate oceanic waters and typically crawl across substrates or migrate by passive drifting, either by adhering to floating objects, or to the underside of the water supported by surface tension [18]. Placozoans can also spread by the asexual reproduction of planktonic 'swarmers' [10,19,20].

Placozoans are commonly found in tropical/sub-tropical regions around the globe with the majority of currently known species being located between 48°N and 35°S [1,21,22]. So far, 22 different haplotypes have been identified using 16S ribosomal sequencing, and the number continues to grow [1,22]. It has previously been shown that placozoans are highly sensitive to temperature changes and have a preference to remain in a stable climate [23]. Moreover, animals that have been isolated from a certain region show a clear preference for the typical conditions in that region [24]. Even though the majority of placozoans have been isolated in tropical regions, there are also cases of placozoans tolerating the cool waters of the Northern Atlantic Ocean (France) and the North Pacific Ocean (Japan) [25,26]. Much fewer biological records (i.e., the temporal biogeographical location of a species) have been available for the Southern Hemisphere and yet not a single record for a Placozoan species has been reported from south of 30 degrees latitude, e.g., south of Melbourne, Australia.

Victoria, a state of Australia, is in a temperate climate zone located in the Southern Hemisphere at a latitude of 37.8°S [27]. A large proportion of the coastline of Victoria is sandy beaches combined with large, shallow rock outcrops. Most of this coastline is part of the Bass Straight in the Tasman Sea and the Indian Ocean with water temperatures ranging from 11.6°C to 20.2°C [28]. In Australia, the oceanic water temperatures peak in the summer/autumn months between November and April, whereas the colder temperatures are experienced in August in the final month of winter [28]. There are no previously reported cases of placozoans being found in Victoria, however the H2 strain has been predicted to tolerate harsher colder climates [23]. Placozoan samples have been isolated before from more northern Australian waters, including the tropical regions of Queensland and the temperate region of South Australia [1].

Here we report the most southern biological records of placozoans, found at a latitude of 38°South in the southern waters of the Indian Ocean in Victoria, Australia. The new records support the growing body of evidence that placozoans can survive in colder climates than previously thought. We compare the Melbourne placozoan strains morphology with previously reported placozoans and importantly, identify a new haplotype (H20) which defines a new clade of Placozoa.

## Materials and methods

### Culturing of placozoans

Placozoans were cultured in 15 cm glass petri-dishes containing artificial sea water (ASW; Instant Ocean, Aquarium Systems, Canada) that was maintained between 3.4% and 4.0% salinity. ASW was refreshed every 2 weeks or if the salinity reached 4.0% via the removal of 50% of the seawater and replacing it with 3.5% ASW. Plates were stored at room temperature

(18–22°C) in indirect sunlight and by means of light strips kept on a constant light cycle consisting of 16 hours of light and 8 hours of dark (LD 16:8). Animals were first kept on their algae-covered slides collected from the ocean (see below). Once isolated, animals were either fed on fresh algae-covered ocean slides, or directly on *Rhodomonas salina* (CSIRO, Hobart, Australia) by renewing the food every 1–2 weeks.

### Trap sampling of placozoans

Slide traps were designed which could be easily placed and collected regularly by a scuba diving team. Small, clear plastic slide mailer-boxes had their sides cut out and glass slides were placed within the box, which were then securely latched closed. The Dive La Trobe scuba diving teams suspended traps from the side of piers or pillars at a depth of 3–5 meters below low-tide level. Trap collection and replacement occurred every 4–6 weeks, where collected traps were placed in large clear plastic 2L bottles (Nalgene, Rochester, USA) for transport to the lab. Here slides were moved to 15 cm Glass petri-dishes filled with a 50:50 mix of artificial seawater (Instant Ocean, Aquarium Systems, Canada) and seawater collected from the collection site. Samples were examined regularly over a period of 8 weeks for the presence of placozoans. Animals were collected under works permit 06.20V1 issued by Parks Victoria, Melbourne, Australia.

### Rock sampling of placozoans

Rocks of 1–5 cm in diameter were collected from rockpools, under piers and in sheltered oceanic waters around Port Phillip Bay, Melbourne at depths of 20 cm to 80 cm. Around 10–20 rocks were placed for transport to the lab in small, sealed containers containing seawater from the sampling location. Samples were examined as well as glass containers as per slide sampling above.

### Placozoan clonal isolation and preparation

When placozoans were found, single animals were isolated and placed on a fresh algae-covered slide collected from the same oceanic location as the animals were isolated from. The slides were soaked in sterile MilliQ water for at least an hour to destroy any diatoms, placozoans or other animals that may have been present on oceanic algae slides. These slides were moved to a new 15 cm glass plate where the single animal isolates were added on top of the slide. Here animals divided by fission creating clonal cultures. Once the number of animals exceeded 10, one animal was taken for strain or haplotype identification by means of PCR (see below).

### Genotyping

Single animals were placed in a 1.5 mL Eppendorf tube with 10 μL QuickExtract DNA Extraction Solution (Biosearch Technologies, Hoddesdon, UK) to provide the cell extract. DNA-extraction followed a standard protocol [29] and samples were stored at −20°C. Polymerase chain reaction (PCR) with placozoan specific primers was performed to identify the placozoan species. The following primers were used: Universal Placozoan primers that span the 3' end of the nuclear small sub-unit (SSU) and the 5' end of the nuclear large sub-unit (LSU): Forward 5′GGTTTCCGTAGGTGAACCTGCGGAAGGATC3; reverse 5′GCATATCAATAAGCGGAGGA3′ were used to confirm placozoan identity as reported by [30]. Then, 16S Forward 5'CGAGAAGACCCCATTGAGCTTTACTA3' and 16S Reverse 5'TACGCTGTTATCCCCATGGTAACTTT3' were used to identify and distinguish the placozoan species based on size of amplicon [30]. PCR reactions were prepared as per Q5 DNA Polymerase PCR

protocol (Q5 High-Fidelity DNA polymerase; New England Biolabs, USA). A 1:10 dilution was made of each DNA sample and of this 2 µL of template DNA was used per reaction using the following cycling conditions: 95 °C denaturation for 2 min; 5 cycles of 95°C for 30 seconds, 63°C for 30 seconds and 72°C for 1 minute; then 20 cycles of 95 °C for 30 seconds, 61°C for 30 seconds and 72 °C for 1 minute with a final extension of 72 °C for 10 minutes (T100 Thermo-cycler, Bio-Rad, California, USA). Products were visualised on 1% agarose gels and saved for DNA sequencing.

### Sequencing and bioinformatic analysis

Before sequencing ExoSAP-IT (Applied Biosystems, Waltham, Massachusetts, USA) was added to the PCR products to remove any unused primer and nucleotides according to the manufacturer's protocol. Sanger sequencing was conducted by the Australian Genome Research Facility (AGRF; Melbourne, Australia) using the 16S rDNA Forward or Reverse primer. Only sequences with high signal peaks were included and aligned to known hap-lotypes [14,16,30] using the multiple sequence alignments tool and Jalview software [31]. Sequences were replicated to achieved high peak signal and concatenated to form a 479 bp sequence that was deposited in GenBank (Accession: PQ614183).

### Neighbour Joining Phylogram

Placozoan 16S sequences were used to produce a Neighbour Joining (NJ) Phylogram using MAFFT (Multiple Alignment using Fast Fourier Transform) with an ungapped alignment using the software Mafft v.7 server [32]. Results were prepared in FigTree (v1.4.4) and clades were allocated according to [22] and [1].

### Electron microscopy

For sample preparation, animals were washed in 3.5% ASW and left to adhere to the base of a small glass dish. Most of the ASW was slowly removed, keeping the animal slightly covered, before fixative was added to the wells (2.5% Glutaraldehyde and 1% Osmium tetroxide in 0.1M Sorrensen's phosphate buffer). Once chemically fixed, animals were dehydrated in a graded ethanol series (30, 50, 70, 80, 90, 100%) in a pelco biowave for 2 minutes at 150 W. Spurr's resin was infiltrated with increasing concentrations (1:2, 1:1, 2:1) of resin:ethanol, then three washes in 100% resin before samples were left to cure overnight at 60 degrees. For Transmission Electron Microscopy (TEM), one block face was prepared. Using an ultramicrotome (Leica, Wetzlar, Germany) sections 100 nm thick were prepared and were picked up on 100 mesh thin bar grids (Proscitech, Townsville, Australia) before being imaged in the TEM (Jeol JEM-2100, Tokyo, Japan) with an AMT NanoSprint 15 II camera (Newspec, Adelaide, Australia).

### Live microscopy

Animals were washed twice in Artificial Sea Water (ASW) before being placed on a slide/plate for imaging analysis. Animals were analysed using a Brightfield upright inverted microscope (Olympus IX81, Tokyo, Japan) with a 10× or 20× inverted lens to examine ultrastructure, movement and behaviour. A Brightfield stereoscope microscope with swan lights was used to examine movement and behaviour (Zeiss, Oberkochen, Germany).

### Confocal microscopy

Animals were washed in 3.5% filtered ASW and placed in a 12-well glass chamber slide containing 3.5% ASW and left overnight to adhere to the base of the glass. For fixation, the

protocol from [33] was modified. Briefly, keeping the animal submerged, all extra ASW was removed before ice-cold 4% PFA and 0.2% Glutaraldehyde in 1% NaCl ASW was added to the animals and left for 2 minutes.

Solution was removed and replaced with ice-cold 4% PFA in High NaCl ASW and left in fixative for 1 hour at 4°C. Post-fixation, animals were washed 4 times with ice-cold PBS before proceeding with immunofluorescent staining.

## Immunofluorescence analysis

Post-fixation, animals were permeabilised with 0.3% Triton-X/PBS before being blocked with 2% BSA (Scientifix, Clayton, Australia) in PBS solution. Primary antibodies α-Actin clone JLA20 (Merck, NJ, USA) and α-panMAGUK clone K28/86 (Merck, NJ, USA) were diluted in 2% BSA/PBS and left overnight at 4°C. Animals were washed 4 times with 0.05% Tween20 (Sigma Aldrich, St. Louis, USA) in PBS (PBST) before secondary antibodies were diluted in 2% BSA/PBS (Alexa Fluor donkey anti-mouse 568 (Ab175472; Abcam, Cambridge, UK)) and Alexa Fluor Goat anti-rabbit 633 (A21071; Abcam, Cambridge, UK) and left for 2 hours covered from light. Animals were washed with PBST twice then once with PBS before 2 μg/mL Hoechst (Abcam, Cambridge, UK) was applied for 30 minutes. Samples were transferred to glass imaging slides and sealed with ProLong glass antifade mount (Invitrogen, California, USA) and left for 24 hours, covered from light, at room temperature before being imaged. Slides were stored covered from light in 4°C for up to 2 months.

## Confocal imaging

Zeiss confocal LSM 800 (Zeiss, Oberkochen, Germany) was utilised to visualise fixed samples under oil immersion using x40 or x63 lens (Zeiss, Oberkochen, Germany) and Zen software before data was analysed using FIJI suite.

## Results

### Isolation of placozoans from the southern waters of Australia

A total of 12 sites were sampled using rock or trap sampling, with placozoans successfully being isolated from three independent sites in Victoria, Australia. Placozoan sample sites included two open water dive sites and one pier located at Blairgowrie on the peninsula of Port Phillip Bay, Victoria, Australia (Fig 1a–1c, S1 Fig). Due to the high tidal movement along the coast of Victoria, the collection and placement of traps was limited to waters protected from breaking waves (e.g., behind a seawall as is found in a marina). Since early 2020, we regularly collected placozoans from Blairgowrie Pier (38.35°S, 144.77°E), which contains a seawall located on the southern peninsula of Port Phillip Bay (Fig 1c). The sites are at the side of a pier rich with corals, plants, fish, and other marine life. The salinity remained constant over the years, between 3.5% - 3.6%, while the water temperature ranged between 13 – 20°C (Fig 1d). The presence of placozoans in Victorian waters validates the ecological niche predicted by Paknia and Schierwater (2015) and their proposition that the animals have the capacity to tolerate cooler climates. Furthermore, the isolation of placozoans from Blairgowrie, Victoria, represents the most southern record of placozoans yet.

### A new placozoan haplotype

A total of 49 placozoans were isolated for genotyping. Using the 16S rDNA fragment, 39 isolates were confirmed as *Trichoplax* sp. H2 (haplotype H2, family Trichoplacidae, genus *Trichoplax*) [22,30,35] with 100% sequence identity to the known *Trichoplax* sp. H2 sequence

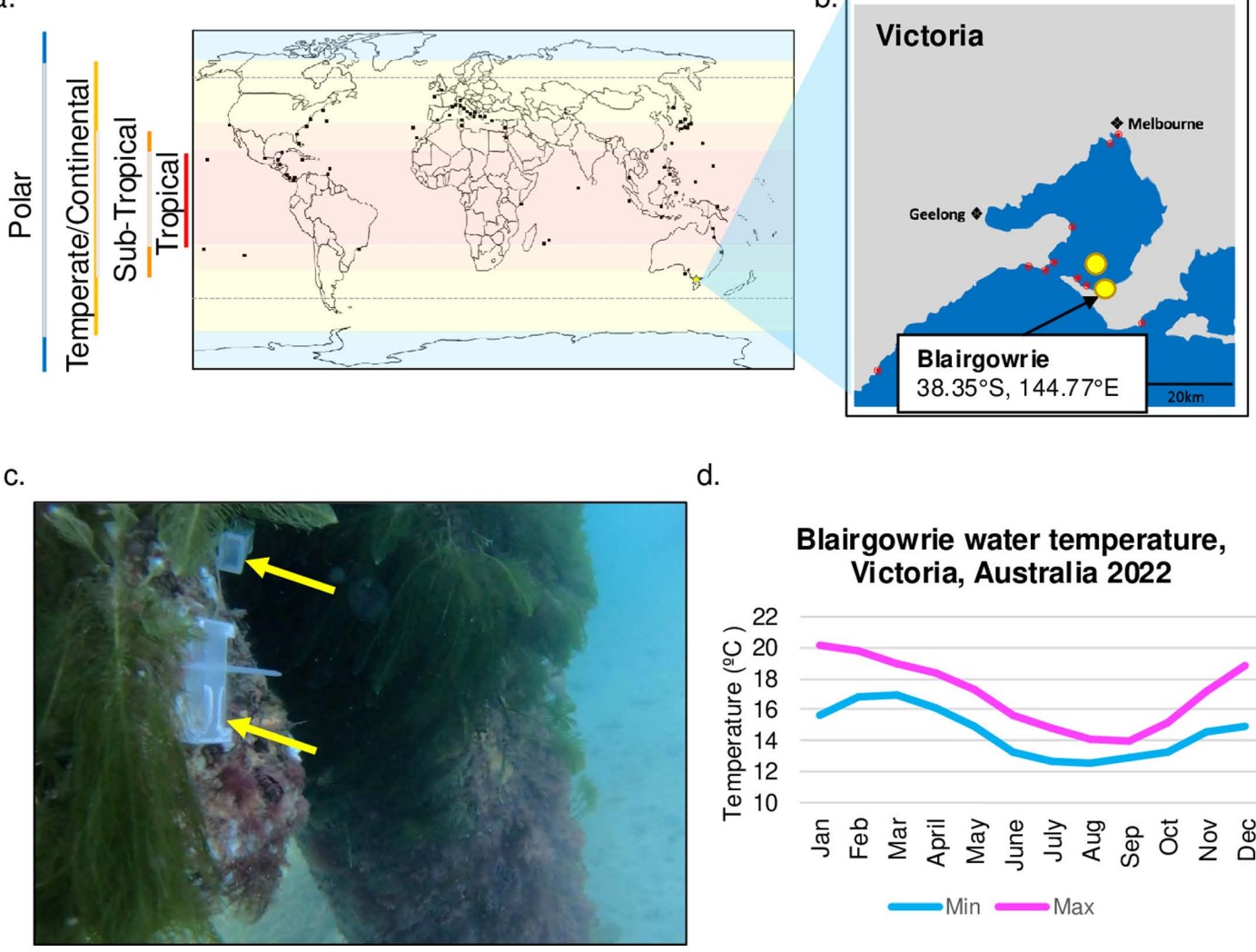

**Fig 1. Isolation of placozoans for Southern waters of Australia. a.** Black squares represent the areas where placozoans have been isolated and the shading delineates the climactic zones from [27,34]. Grey dotted line is the predicted location of placozoans based on [23]. This drawing is for illustrative purposes only. **b.** is an enlarged section of a. showing the location of collection points in Victoria around Port Phillip Bay (red dots) including sites where placozoans were isolated (yellow dots). This drawing is for illustrative purposes only. **c.** Local environment of the traps placed at Blairgowrie pier where Placozoa were isolated. Each trap can be secured to the side of pillars at a depth of 5 meters. GoPro images captured by MR in 2022 **d.** Water temperatures in the sampling area in 2022 (Blairgowrie, (max/min ºC), Victoria, Australia with water temperatures reaching a maximum of 20°C for one month, whereas most of the year the temperature remains under 17°C and as low as 12°C [28].

(Fig 2b, S2a Fig). 8 samples were identified as *Trichoplax* sp. H17, a sister haplotype to H2, with 100% sequence identity to previously reported H17 strains (S2a Fig) [30]. Both the H17 and H2 strain were predicted to inhabit cooler climates and our findings confirm these predictions. We have seen that H2 is the dominant haplotype in the most southern reporting of any placozoans.

Surprisingly, two placozoan samples produced a much shorter 16S PCR fragment than expected (visually around 350 bp, S2b Fig). To determine if these sequences were known haplotypes, 16S rDNA was used, which contains a highly variable region specific to different *Placozoan* haplotypes [30]. When these sequences were compared to the diagnostic (highly variable) region of the16S rDNA sequence of published placozoan haplotypes [14,16,30] no identical

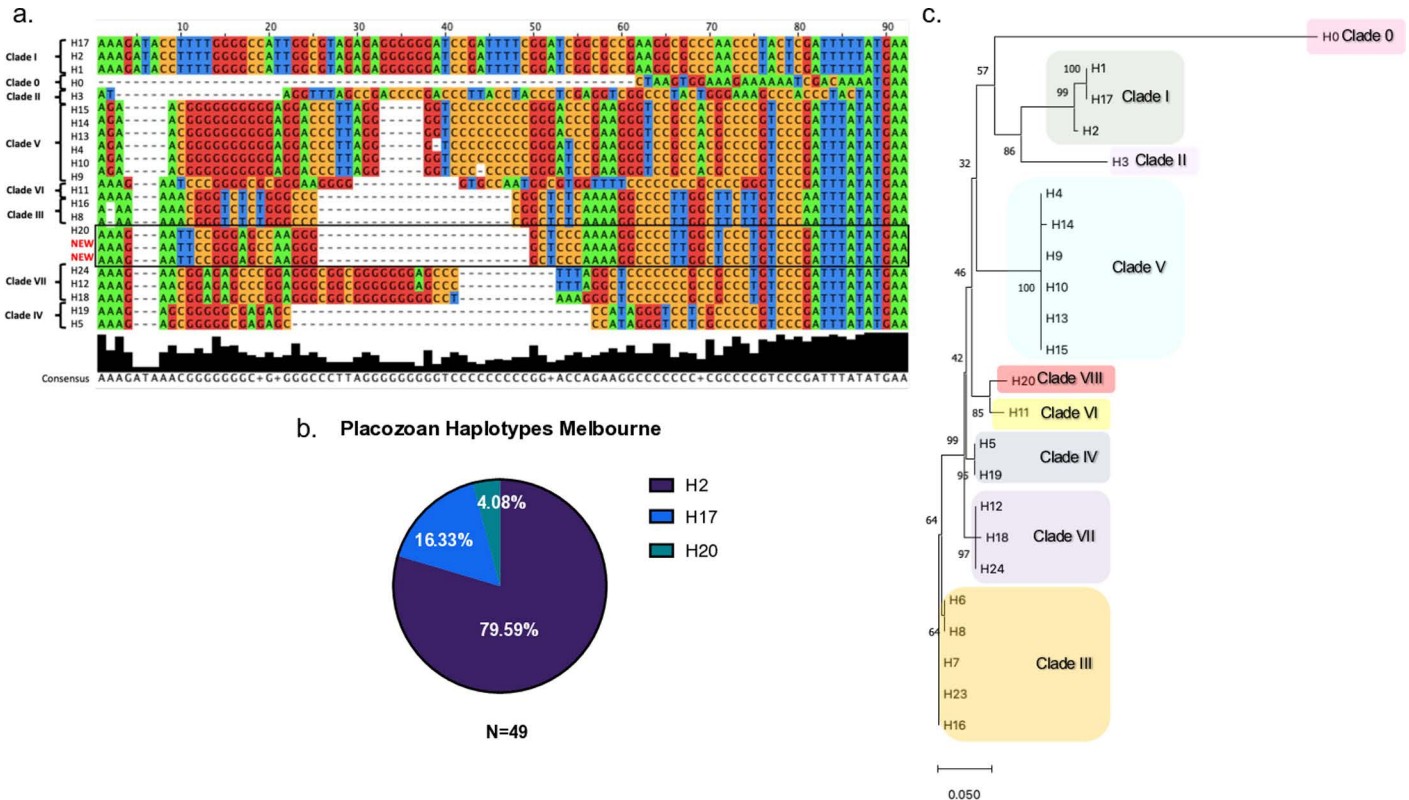

**Fig 2. H20, a new placozoan haplotype found in cooler climate of the Southern waters of Australia. a.** 16S rDNA sequences reveal differences between haplotypes and clades, including the newly isolated H20. Sequences were sourced from NCBI database (see materials and methods), aligned using Clustal Omega and analysed in JalView. **b.** 49 placozoan samples have been sequenced using 16S rDNA revealing Trichoplax Sp. H2 as the dominant haplotype (79.59%) compared with H17 (16.33%) and H20 (4.08%) **c.** A Neighbour Joining (NJ) tree was constructed using 16s rDNA sequences to compare the divergence that is present between different haplotypes.

sequences were found. We have designated the new haplotype as H20 (Fig 2a, 2b). It should be noted that this novel haplotype corresponds to placozoans initially found by M. Eitel in Hong Kong in 2013 (Eitel et al. [unpublished]). The H20 animals isolated at Blairgowrie Pier were collected in the autumn season from two independent sites. They were harvested from trap samples with the water temperature ranging between 16.1°C – 18.4°C during this time. To determine the clade of the new H20 sequence a Neighbour Joining (NJ) tree was constructed. The topology of the NJ tree allowed for the H20 to be comparably slotted into previously published placozoan clade trees. The 16S rDNA Neighbour Joining (NJ) tree showed a 77.98% divergence of the H20 haplotype from already known haplotypes (S1 Table). When clade branch distances (or divergence) were compared with known clades III and IV, the distance values were comparable to that of H20. This data supports the formation of a new novel clade based on the divergence, that will be designated Clade VIII (Fig 2c). In general, the neighbour joining tree of the ungapped sequences matches previously published maximum likelihood trees [22].

Of note, the neighbour joining tree presented here is not intended to represent an accurate phylogenetic reconstruction, rather to provide insight into the relative distances between haplotypes. As seen in other studies, using 16S data only is sufficient for the identification of a new clade. Once whole genome sequencing can be completed, the H20 haplotype can be subjected to maximum likelihood (ML) or Bayesian analyses and learn more about the genetic variation between haplotypes and construct genuine phylogenetic trees [36].

In summary, the new record is the furthest latitudinally (in both a southerly and northerly direction) reporting of a strain that is not *Trichoplax* sp. H2. As the only other H20 isolate is from Hong Kong, the ecological niche and worldwide distribution of H20 remains unknown.

## Morphology and behaviour of placozoans from Victorian waters

We next wanted to investigate the morphology of the newly found placozoan strains using different microscopic techniques. Most placozoans are identical with respect to their flat discoid shape (See Fig 3a) with the exception of the H0 haplotype, *Polyplacotoma mediterranea* [16] which appears elongated with many extensions protruding off the main body. Melbourne placozoans in the lab migrated towards well-lit areas around the edge of the dish. Of note they tended to avoid remaining in contact with other placozoans, unlike the social contact that has been reported previously [37]. As with other placozoan isolates reported by Schierwater et al. [unpublished], the animals demonstrated seasonal changes where their population growth slowed during the winter months, despite constant laboratory conditions. Morphologically, placozoans take on a variety of discoid and distorted shapes [38]. The Melbourne placozoans (H2 and H17) are no different; they can contract to 0.5 mm and elongate to 2–3 mm

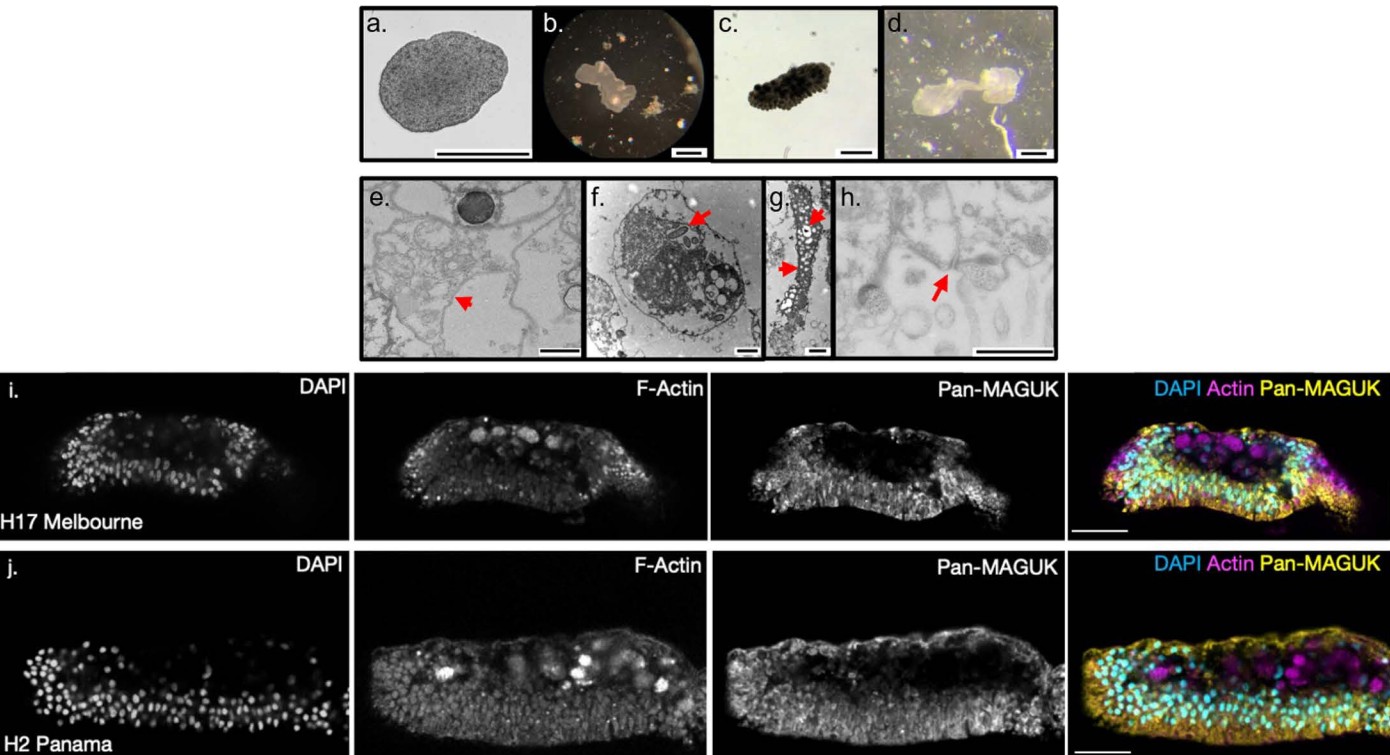

**Fig 3. Macroscopic, microscopic, and ultrastructural characterisation of placozoa isolated in Melbourne, Australia. a.** Inverted phase contrast image of isolated Melbourne placozoa; **b.** Brightfield image of motionless animal that has crawled over particulate matter showing feeding behaviour. **c.** Colour inverted image of a Melbourne placozoan showing multiple punctate. **d.** Brightfield image of Melbourne placozoan undergoing fission. Scale bars represent 500 μm. Transmission Electron Microscope (TEM) images of Melbourne H2 strain (e-h); **e.** cross section through a fibre cell; **f.** Fibre cell is hosting symbiotic bacteria (arrow). **g.** gland cells containing granules (arrows) required for digestion of food and particulate matter. **h.** Adherens junctions in lower epithelium (red arrow) Scale bars represent 1 μm. Immunofluorescence (IF) imaging reveals the same tissue architecture of the **i.** *Trichoplax* sp. H17 Melbourne, and **j.** *Trichoplax* sp. H2 Panama. H17 and H2 show the whole animal architecture. Both illustrate a tri-layered structure that is highly populated on the lower surface and a thin upper layer that is highlighted by Pan-MAGUK antibody (yellow) [39], The internal fibre cells can be visualised by the presence of actin (Magenta) and the nucleus via DAPI stain (Cyan). Scale bars represent 10 μm. Images captured using Confocal LSM 800 with a x63 oil-immersion lens.

depending on whether they are in motion or resting (Fig 3b–3d , S3 Fig ). Similar to other placozoans, the Melbourne H2 and H17 placozoans show a motion behaviour where they traverse a dish in horizontal and vertical movement up the side of the dish; for this they show a combination of rotation, folding and curling movements.

As previously documented, placozoans have nine different somatic cell types with some sub-variants in types identified, with epithelial cells being most prominent on the upper and lower epithelium, interspersed with secretory cells and fibre cells that traverse the middle of the animal [4,13,40,41]. To examine their tissue and cell composition, we compared the *Trichoplax* sp. H2 Melbourne haplotype with existing data on to the H1 strain, as no H2 or H17 haplotype immunofluorescent or Transmission Electron Microscopy (TEM) data have been published. Using TEM imaging, fibre cells and symbiotic bacteria were identified in *Trichoplax* sp. H2 Melbourne (Fig 3e, 3f), which is consistent with previous reports [9,42,43]. Secretory cells were identified in the lower epithelium and around the margins, some of which are responsible for external digestion via the secretion of granulated enzymes (Fig 3g). The presence of junctions are limited in basal metazoans with placozoans only containing adherens junctions [39]. These junctions were present in the hyper intense regions at the apical junction of the lower epithelium in *Trichoplax* sp. H2 Melbourne, which is consistent with adherens junction observations in *Trichoplax adhaerens* H1 (Fig 3h). By immunofluorescent staining we found that the animals contained a structurally dense lower epithelium and a thinner upper epithelium (Fig 3i, 3j). H2 Melbourne shows Actin localised to the apical borders of the upper epithelium and to fibre cells through the middle of the animal, which is consistent with a previously reported analysis of *Trichoplax adhaerens* H1 haplotype [13]. Also, similar to H1 is the localisation of a cell polarity marker for Pan-MAGUK to the lateral borders of the upper epithelium. The localisation to the lateral borders was also observed in the lower epithelium (Fig 3i, 3j).

In summary, there are no obvious morphological differences between the Grell H1 strain and these newly isolated Melbourne H2 strain, neither at a macroscopic nor microscopic level.

## Discussion

Despite substantial sampling efforts for placozoans in the field, our knowledge on the overall placozoan biodiversity and global distribution of placozoan genetic lineages and species is still limited [1,23,24]. Here, we report the first placozoan samples from Melbourne, Australia. These samples are the furthest southerly reporting of placozoans, which illustrates their ability to thrive in a temperate climate. Previous searches for placozoans in the Antarctic ended empty-handed [44]. This research extends the boundaries of the current known haplotype locations and supports the growing evidence that Clade I of placozoans has a significant capacity to adapt to a variety of environments and represents the most robust strain of placozoans. As modelled by Paknia and Schierwater (2015), it was predicted that *Trichoplax* sp. H2 and *Trichoplax* sp. H17 had the capacity to survive in the marine habitat of Melbourne, Australia. However, the discovery of the genetically highly divergent strain Placozoa sp. H20 in the same habitat highlights that even distantly related placozoans share similar ecological niches. In addition, we can confirm the overall morphological similarities between the Melbourne placozoan strains *Trichoplax* sp. H2 and *Trichoplax* sp. H17 with previously reported placozoan strains, such as *Trichoplax* Grell strain, sp. H1, including their tri-layered structure, presence of major cell types such as fibre cells and gland cells, and their highly ciliated lower epithelium.

*Trichoplax* sp. H2 has been reported as the most robust species of placozoans. It is considered to be cosmopolitan, that is, it has the capacity to live and survive in a variety of different environments, including high and low salinities, high and low temperatures, and on a variety

of different substrates (e.g., mangroves, rocks, glass) [1,23]. The capacity of this strain to have a wider tolerance for environmental factors is valuable in the context of usage as a model organism. The most northerly record of a placozoan strain is the report of the H2 strain (Clade I) from Roscoff, France (48.7°N) [25]. Roscoff has similar average water temperatures as Victoria, but a colder minimum, as low as 9.8°C in January [28]. It has been reported that isolates of placozoan strains with a wide distribution range nevertheless show higher population growth at the temperature of their geographic sampling location [24]. For example, *Trichoplax* sp. H2 "Roscoff" that was collected in cooler oceanic waters grew more at cooler water temperatures in the laboratory, rather than at higher water temperatures [24]. The opposite is true for a tropical H2 strain *Trichoplax* sp. H2 "Panama". When placed in cooler water temperatures, the animals failed to thrive, and population numbers decreased [24]. The "Melbourne" H2 and "Roscoff" H2 strain show the same preference. The "Melbourne" H2 can grow and thrive in a laboratory setting with water temperatures between 17–20°C. The predicted modelling [12,23] suggested that placozoans have the capacity to live in environments with water temperatures around 10°C. Our findings confirm and extend these observations.

The isolation of the new Placozoa sp. H20 haplotype was unexpected and the sequence has been deposited in GenBank (accession: PQ514183). This new haplotype has been designated H20 based on the unpublished sampling and isolation of the same species found by M. Eitel in 2013 ([1], Eitel et al. [Unpublished]). Placozoa sp. H20 animals that were found in Hong Kong were sourced from rock collections in August 2013 and have a 100% 16S rDNA sequence identity to H20 found in Australia (Eitel et at. [Unpublished]). The sea temperature in Hong Kong is around 28°C in the summer, which is vastly different to the conditions observed in Melbourne. The second isolation of this haplotype in a substantially different environment suggests that H20 is another robust species with a broader ecologic niche. In general, different clades are often located in different climatic areas around the globe [1] with the highest diversity of range latitude is seen in Clade I. There are some examples of Clade V and IV being located further north in latitude. For example, a sample from Clade IV isolated from Italy was at a latitude of 41°N where the water temperatures range between 14°C – 25°C [1]. Previously completed modelling predicted that Clade I (including the H2 haplotype) could survive in an ecological habitat as far south as 44°S [23]. However, other Clades such as Clades III and IV are predicted based on location data to be much more stenoecious, i.e., to only be able to live in a restrictive range of habitats [23]. Further sampling over wider ranges of different environmental conditions is needed to better understand the ecological niches of placozoan strains.

Our work extends the understanding of the ecology and diversity of Placozoa. It demonstrates the tolerance of placozoans for cooler climates and provides the first morphological and ultrastructural characterisation of placozoan clades isolated from the Southern waters of Australia, adding to the growing ultrastructural characterisation of placozoan strains. The data suggest further sampling in temperate marine environments to further improve our understanding of the global distribution and the evolution of placozoans.

## Supporting information

**S1 Fig. Details of Placozoan traps and their placements. a.** Schematic of the 5-slide mailer boxes for glass slides that are attached to the piers using cable ties. **b. and c.** Attachment of traps are to the side of piers. Traps are placed and collected by a scuba diving team under the piers on pylons. A main cable is secured around the pylon and is used as the main attachment point and easy removal without removing all traps at once if need be. Traps are maintained there for a period of 4–6 weeks at a depth around 5 meters depth well under tidal mark before collection.
(PDF)

**S2 Fig. Sequence alignment collected of Melbourne H2 and H17 samples. a.** Sequence alignment using 16S rDNA fragment show the two species in Clade I – H2 (Panama) and H17 (Purple panels). H17 and H2 can be distinguished by single base pair variations and a 13bp gap in the 16S rDNA sequence. All sampled H2 and H17 haplotypes from Melbourne that were sequenced with 16S rDNA revealed identical sequences with the previously documented H2 and H17 species respectively. **b.** 16S rDNA fragment PCR products revealed two bands that were shorter in length compared with the other placozoan sequences indicating potential new haplotype H20. Each lane represents rDNA from a single placozoan sample with the following haplotypes identified: PH1041 – H2, PH1043 – H2, PH1045 – H2, PH1047 – H2, PH1051 – H2, PH1053 – H2, PH1037 - H20, PH1055 – H2, PH1059 -H2, PH1064 – H2, PH1062 – H2, PH1063 – H2, PH1061 – H2, PH1039 – H20, PH1065- H2, PH1066 – H2.
(PDF)

**S3 Fig. Light microscopy analysis of Melbourne Placozoans.** Brightfield microscopy analysis of Melbourne Placozoans show they have the capacity to take on many different shapes, forms and sizes as observed with many different placozoan haplotypes. Scale bar represents 500 μm.
(PDF)

**S1 Table. 16S Placozoan mafft Data.**
(XLSX)

**S1 Raw Image. Raw Image for Gel in S2b Fig.**
(TIFF)

## Acknowledgments

We would like to acknowledge and thank Parks Victoria for their support in securing permits and the Blairgowrie Yacht Squadron allowing access to areas within the bay. We would like to thank Dive La Trobe for their ongoing support in placing and collecting traps year-round. We want to acknowledge the assistance of the La Trobe University Bioimaging facility for their support in this project. I would like to acknowledge the assistance, correspondence, and ongoing support from M. Eitel in the preparation of this manuscript.

## Author contributions

**Conceptualization:** Bernd Schierwater, Patrick O. Humbert.

**Data curation:** Bree A. Wright, Hans-Jürgen Osigus, Moritz J. Schmidt, Julian Ratcliffe, Kai Kamm.

**Formal analysis:** Hans-Jürgen Osigus, Kai Kamm.

**Methodology:** Bree A. Wright.

**Resources:** Julian Ratcliffe, Gabriela C. Martinez-Ortiz, Martina Rehn.

**Software:** Moritz J. Schmidt, Kai Kamm.

**Validation:** Kai Kamm.

**Writing – original draft:** Bree A. Wright.

**Writing – review & editing:** Bree A. Wright, Hans-Jürgen Osigus, Moritz J. Schmidt, Julian Ratcliffe, Kai Kamm, Gabriela C. Martinez-Ortiz, Martina Rehn, Marc Kvansakul, Bernd Schierwater, Patrick O. Humbert.

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
