## [Decision Letter · Decision Letter 0]

9 Jan 2024

PONE-D-23-35441Tolerance of Placozoa for temperate climates: Evidence for known and new placozoan clades in the southern waters of AustraliaPLOS ONE

Dear Dr. Humbert,

Thank you for submitting your manuscript to PLOS ONE. After careful consideration, we feel that it has merit but does not fully meet PLOS ONE’s publication criteria as it currently stands. Therefore, we invite you to submit a revised version of the manuscript that addresses the points raised during the review process.

We look forward to receiving your revised manuscript.

Kind regards,

Robert E. Steele, Ph.D.

Academic Editor

PLOS ONE

Journal Requirements:

"We would like to acknowledge and thank Parks Victoria for their support in securing permits and the Blairgowrie Yacht Squadron allowing access to areas within the bay. We would like to thank Dive La Trobe for their ongoing support in placing and collecting traps year-round. We want to acknowledge the assistance of the La Trobe University Bioimaging facility for their support in this project. I would like to acknowledge the assistance, correspondence, and ongoing support from M. Eitel in the preparation of this manuscript. This work was funded through a small grant from the La Trobe University Research Focus Area Grant Ready Scheme  #2000004451."

"POH,MK

La Trobe Univerisy Grant Ready Scheme funding

no.2000004451

Funding body had no participation in the preparation or completion of any work."

3. We note that you have referenced 

(M. Eitel, Personal Communication, unpublished, sequence denoted as “New” on Fig 2a), 

(B. Schierwater, unpublished data),  ([1],Personal Communication, unpublished) and  (Eitel, unpublished)

which has currently not yet been accepted for publication. Please remove this from your References and amend this to state in the body of your manuscript: (ie “Bewick et al. [Unpublished]”) as detailed online in our guide for authors

Additional Editor Comments:

I am sorry it has taken so long for this review. I really struggled to find individuals who would agree to review your manuscript, and then I was unable to get them to submit their reviews. I believe that the one review I was able to obtain is appropriate and I concur with this reviewer's comments and recommendation that only minor revision is needed.

Reviewers' comments:

Reviewer's Responses to Questions

**Comments to the Author**

1. Is the manuscript technically sound, and do the data support the conclusions?

Reviewer #1: Yes

2. Has the statistical analysis been performed appropriately and rigorously? 

Reviewer #1: Yes

3. Have the authors made all data underlying the findings in their manuscript fully available?

Reviewer #1: No

4. Is the manuscript presented in an intelligible fashion and written in standard English?

Reviewer #1: Yes

5. Review Comments to the Author

Reviewer #1: This manuscript presents new results relevant to understanding the morphological diversity, biogeography, and ecology of placozoans. Methods and results are appropriate and are presented clearly, and the discussion of relevant literature is thorough and informative. I recommend publication following minor revisions to address the following points.

The 16S sequence obtained for haplotype H20 animals has not been previously reported. It is not clear whether all the unique features of this sequence are documented in Figure 2a. If they are, the manuscript should make that clear. If they are not, the sequence data should be made available through submission to NCBI or in another way.

Lines 133-135. It would be good to mention what location or locations the algae-covered slides came from. Culturing newly collected animals on such slides seems like a potential source of contamination with placozoans from the location of slide collection (perhaps from one of the more northern sites shown in Figure 1b). It would also be good to mention how long the slides were soaked in sterile water. If there is a technique paper describing this approach as part of culturing placozoans, it would be great to cite that paper.

Lines 388-394. It would be clearer to replace “prefer culture conditions” with “show higher population growth rates under culture conditions” and replace “thrived” with “grew more.”

Lines 414-416. It would be clearer to change “are predicted” to “are predicted based on location data” This will make clear that these predictions aren’t based on experimental data.

The listing for reference 38 should be replaced with

https://www.nature.com/articles/s41567-020-01134-7

6. PLOS authors have the option to publish the peer review history of their article (what does this mean? ). If published, this will include your full peer review and any attached files.

**Do you want your identity to be public for this peer review?** For information about this choice, including consent withdrawal, please see our Privacy Policy .

Reviewer #1: No

---

## [Author Response · Author response to Decision Letter 1]

3 Dec 2024

Thank you for providing helpful amendments for the manuscript Tolerance of Placozoa for temperate climates: Evidence for known and new placozoan clades in the southern waters of Australia. All the suggested changes have been made with some additional typographical and clarification additions that arose through the review of this paper in the primary authors PhD submission. A summary of changes is provided in the attached document.

---

## [Editor Report · Decision Letter 1]

8 Jan 2025

Tolerance of Placozoa for temperate climates: Evidence for known and new placozoan clades in the southern waters of Australia

PONE-D-23-35441R1

Dear Dr. Humbert,

We’re pleased to inform you that your manuscript has been judged scientifically suitable for publication and will be formally accepted for publication once it meets all outstanding technical requirements.

Kind regards,

Robert E. Steele, Ph.D.

Academic Editor

PLOS ONE

Additional Editor Comments (optional):

The authors have satisfactorily addressed the issues raised in the initial review.
---

## [Editor Report · Acceptance letter]

PONE-D-23-35441R1

PLOS ONE

Dear Dr. Humbert,

I'm pleased to inform you that your manuscript has been deemed suitable for publication in PLOS ONE. Congratulations! Your manuscript is now being handed over to our production team.

Kind regards,

on behalf of

Dr. Robert E. Steele

Academic Editor

PLOS ONE